# Deep learning-based stress detection for daily life use using single-channel EEG and GSR in a virtual reality interview paradigm

**Hun-gyeom Kim**[1☯], **Solwoong Song**[1☯], **Baek Hwan Cho**[2,3]*, **Dong Pyo Jang**[1]*

**1** Department of Biomedical Engineering, Hanyang University, Seoul, Korea, **2** Department of Biomedical Informatics, CHA University School of Medicine, CHA University, Seongnam, Korea, **3** Institute of Biomedical Informatics, CHA University School of Medicine, CHA University, Seongnam, Korea

☯ These authors contributed equally to this work.
* baekhwan.cho@cha.ac.kr (BHC); dongpjang@hanyang.ac.kr (DPJ)

**Data Availability Statement:** According to the approved IRB, we have shared the participant data in a public repository. All the data acquired during the experiment has been uploaded to the following

## Abstract

This research aims to establish a practical stress detection framework by integrating physiological indicators and deep learning techniques. Utilizing a virtual reality (VR) interview paradigm mirroring real-world scenarios, our focus is on classifying stress states through accessible single-channel electroencephalogram (EEG) and galvanic skin response (GSR) data. Thirty participants underwent stress-inducing VR interviews, with biosignals recorded for deep learning models. Five convolutional neural network (CNN) architectures and one Vision Transformer model, including a multiple-column structure combining EEG and GSR features, showed heightened predictive capabilities and an enhanced area under the receiver operating characteristic curve (AUROC) in stress prediction compared to single-column models. Our experimental protocol effectively elicited stress responses, observed through fluctuations in stress visual analogue scale (VAS), EEG, and GSR metrics. In the single-column architecture, ResNet-152 excelled with a GSR AUROC of 0.944 (±0.027), while the Vision Transformer performed well in EEG, achieving peak AUROC values of 0.886 (±0.069) respectively. Notably, the multiple-column structure, based on ResNet-50, achieved the highest AUROC value of 0.954 (±0.018) in stress classification. Through VR-based simulated interviews, our study induced social stress responses, leading to significant modifications in GSR and EEG measurements. Deep learning models precisely classified stress levels, with the multiple-column strategy demonstrating superiority. Additionally, discreetly placing single-channel EEG measurements behind the ear enhances the convenience and accuracy of stress detection in everyday situations.

## 1. Introduction

The investigation into stress holds a central position within the fields of psychology and healthcare, owing to its profound impact on the overall well-being, encompassing both physical and mental aspects, of individuals. According to Lazarus and Folkman [1], stress is defined as "a

webpage: https://github.com/MIHLabCHA/VR-Stress-Interview-Experiment-Data Please feel free to check it at your convenience.

**Funding:** This work was supported by the Brain Convergence Research Program of the National Research Foundation (NRF) funded by the Korean government (MSIT) (No. NRF-2023R1A2C2003577). This is all the funding that was received for this study.

**Competing interests:** The authors have declared that no competing interests exist.

specific relationship between an individual and their environment, characterized by the individual's appraisal of the situation as exceeding their resources and threatening their well-being." Stress can originate from a variety of sources, including work-related pressures, interpersonal relationships, financial difficulties, and health-related challenges. Prolonged exposure to stress can lead to adverse health outcomes, such as anxiety, depression, cardiovascular disorders, and compromised immune system function [2].

The examination of physiological signals plays a critical role in stress research, offering insights into the complex physiological responses to stressors. Objective metrics, such as heart rate, blood pressure, skin conductance, and cortisol levels, provide valuable validation for self-reported stress assessments and illuminate the underlying biological mechanisms of stress [3]. These physiological signals also enable real-time monitoring of stress levels, offering valuable insights for stress management interventions. For example, biofeedback techniques leverage physiological cues to empower individuals to regulate their stress responses, ultimately improving stress management outcomes [4]. Furthermore, these signals hold the potential to provide insights into the fundamental biological mechanisms of stress, offering real-time monitoring and identifying individuals at higher risk of stress-related disorders.

Commonly used physiological signals in stress detection methodologies include heart rate (HR) (HR) [5], heart rate variability (HRV) [6–8], galvanic skin response (GSR) [9], voice [10], respiration rate (RR) [11], and electroencephalogram (EEG) [12–14]. In recent years, deep learning methods have been employed to analyze physiological signals in the context of stress research. Notable examples include the use of a convolutional neural network (CNN) to distinguish between stress and non-stress states using electrocardiogram (ECG) signals, achieving an impressive accuracy of 92% with 2-second ECG signals [15]. Another study utilized long short-term memory (LSTM) algorithms to categorize stress and non-stress conditions based on HRV, achieving an accuracy exceeding 90% [16]. Additionally, deep learning techniques have been applied to analyze stress levels through EEG signals, achieving an accuracy of over 85% using a 1D-CNN model trained on 1-second EEG signal data [17]. These findings underscore the potential of deep learning in comprehensively interpreting physiological signals in stress research.

While the aforementioned studies highlight the promise of deep learning in analyzing physiological signals in stress research, it is essential to acknowledge inherent limitations in this approach. One such limitation relates to data collection challenges, especially for physiological signals like multi-channel EEG, which often require specialized equipment such as EEG caps or forehead attachments. Furthermore, it is important to recognize the prevalence of frequently used stress-inducing tasks, such as the Stroop task or arithmetic exercises, in existing literature [18, 19]. However, these tasks may not fully replicate the stressors encountered in daily life, and their controlled nature could introduce artificial elements, potentially overlooking the diverse range of stresses individuals face in their everyday routines. This consideration gains particular significance when viewed in the context of real-world applications.

In light of the aforementioned factors, the present study utilized a VR-based mock interview environment to offer a more ecologically valid approach to inducing stress, effectively simulating real-life scenarios that individuals might encounter. This stands in contrast to prior research that relied on standardized laboratory-based stress induction techniques, which might not authentically replicate real-world stressors. Additionally, the study embraced an easily applicable methodology for acquiring 1-channel EEG and GSR signals. The implementation of the 1-channel EEG and GSR signal acquisition methodology, as employed in this study, effectively addresses these concerns. Lastly, the study harnessed deep learning algorithms for stress classification, demonstrating promise in accurately discerning stress levels from physiological signals.

## 2. Methods

### 2.1 Experiment design

This study conducted its participant recruitment from February 1, 2022, to September 1, 2022, and obtained written consent from participants for their involvement in the experiment. This consent included agreement for the research team to have direct access to their personal and acquired data. We enrolled 30 healthy young adults aged 20, comprising 16 males and females, with a mean age of 23.63 ± 3.03, all devoid of neurological or cardiovascular disorders. Participants were instructed to abstain from nicotine, alcohol, and caffeine consumption for a day prior to the study. Ethical approval for the research protocol was obtained from Hanyang University's Institutional Bioethics Committee (IRB: HYUIRB-202201-016).

As illustrated in Fig 1(A), participants were equipped with a behind-the-ear (BTE) EEG device as described in reference [20]. The placement of electrodes was determined following established protocols from prior foundational studies [21]. This experimental configuration employed the BTE EEG apparatus to record EEG signals from participants while they were in seated position.

Additionally, instrumentation from Biopac System Inc., USA, was employed to monitor participants' Galvanic Skin Response (GSR). Participants rated their stress levels on a scale from 0 to 10, with decimal places to the first digit, ranging from "No stress" to "Severe stress" [22]. Biosignal measurements were recorded continuously from the initiation of the experimental paradigm.

### 2.2 Biosignal processing

To validate the induction of stress within this experimental paradigm, classical signal processing techniques were applied to the recorded biosignals, and the results were compared with established biosignal indicators associated with stress. The analysis was conducted using

(a)

(b)

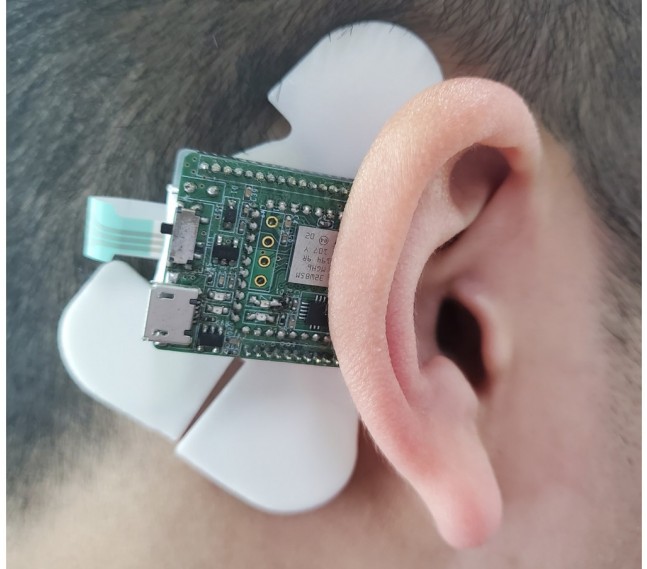
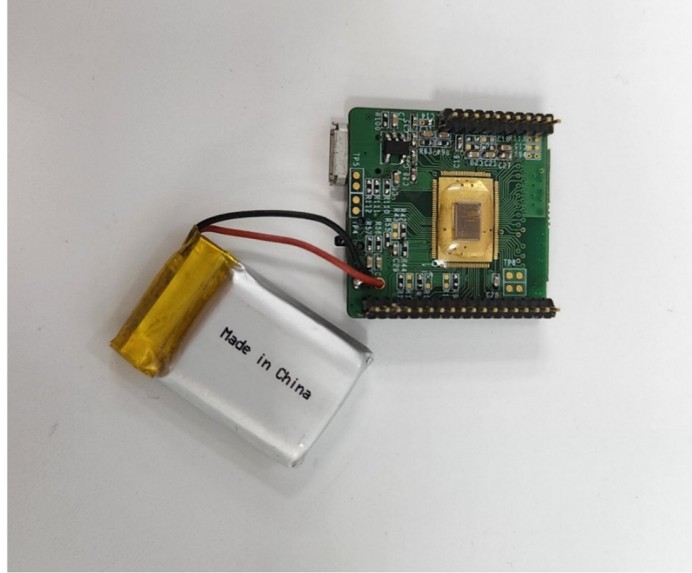

**Fig 1.** Electrode placements: The electrode locations were determined based on prior foundational investigations, designating the superior auricular region for the channel, a transitional position for the ground, and the mastoid area as the reference point (a) Application of the measurement device (b) BTE EEG measurement device.

MATLAB (The MathWorks Inc., 2022), in conjunction with its EEGLAB and LEDALAB tool-boxes [23, 24]. The EEG data underwent bandpass filtering in the range of 1Hz to 50Hz, and power within the alpha (8-12Hz) and beta (15-30Hz) frequency bands was computed. Furthermore, the GSR data were downsampled to 10Hz, followed by a decomposition process into tonic components, representing gradual changes, and phasic components, reflecting rapid variations, using Continuous Decomposition analysis. Subsequently, the magnitude of each component was quantified.

## 2.3 Dataset and preprocessing

Experiments involved a total of 30 participants; however, due to noise concerns, data from 29 participants were subjected to analysis. The GSR frequently exhibits two distinct components: the tonic and phasic responses. The tonic component represents the slower-changing aspect, whereas the phasic component captures rapid fluctuations. These components typically emerge within the frequency range of 1 to 15Hz. In contrast, exploration of EEG signals often reveals a broader frequency spectrum, spanning from 1 to 30Hz. Appropriate filtering was applied to address this. For subsequent AI analysis using both CNN and transformer-based models, data were segmented into 30-second intervals with a 50% overlap and transformed into grayscale spectrogram format with dimensions of 64x64 pixels. Following these steps, a dataset comprising 551 samples of the Normal condition and 1529 samples of the Stress condition was compiled. Furthermore, a cost-sensitive learning approach [25] was employed to address class imbalance, assigning greater weight to the minority class without altering the total number of data instances.

## 2.4 CNN and transformer model architectures

In this study, five distinct CNN models one Vision Transformer model were employed as foundational structures [26]. The models used ResNet-50 [27], ResNet-152, EfficientNet-b0 [28], DenseNet-161 [29], and Inception-v3 [30], alongside the non-CNN Vision Transformer. Both single-column and multiple-column configurations, detailed in the following sections, were adopted for binary classification of stress states using EEG and GSR data. The single-column approach involved training the model using either EEG or GSR data. Each model was initialized using a pre-trained ImageNet model due to significantly lower performance when training from scratch (data not shown). As depicted in Fig 2, the model comprised a model-specific backbone(either CNN or transformer-based), a fully connected layer, a dropout layer, and a ReLU activation function. For optimization, the Adam optimizer was employed with an initial learning rate of 0.001, and the training epoch was set to 30. Additionally, a batch size of 32 was utilized. In addition, to leverage information from both EEG and GSR modalities concurrently and jointly extract features, a multiple-column configuration was devised. GSR and EEG features were extracted from each model backbone and subsequently concatenated. For normalization, mean subtraction was applied to each pixel value in the spectrogram image to balance the contribution of each modality, preventing any one from disproportionately influencing the model's performance. A fully connected layer, accompanied by a dropout layer and a ReLU activation function, followed this concatenation [31, 32]. To enhance training efficiency, weights exhibiting optimal performance in the single-column models were employed for initialization. The multiple-column model utilized the Adam optimizer with an initial learning rate of 0.0001, a training epoch of 30, and a batch size of 32. In contrast to the approach used with the single-column model, this specific learning rate was selected because the model had already been partially trained, necessitating careful fine-tuning during updates to optimize performance effectively. Furthermore, when comparing the multiple-column

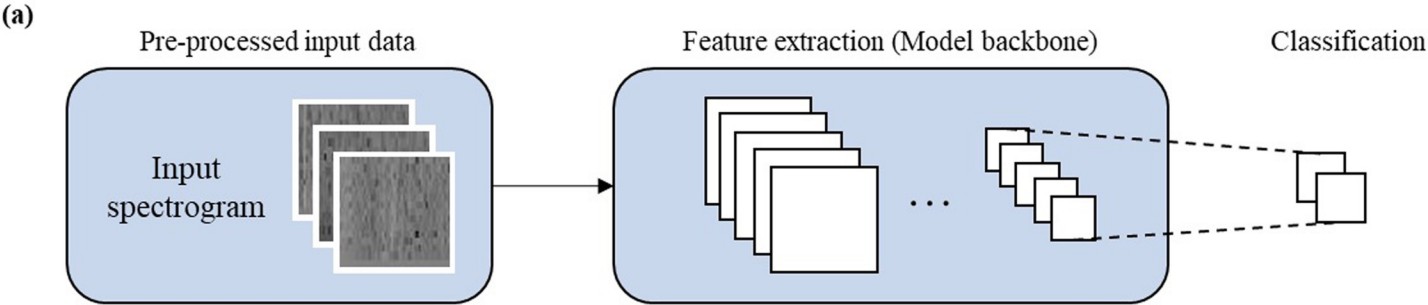

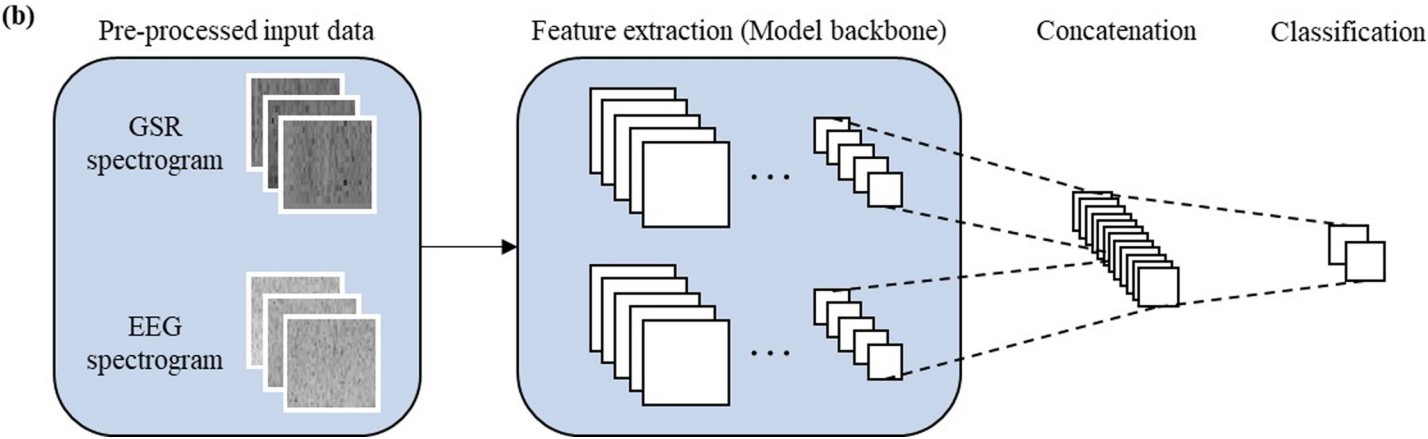

**Fig 2.** Overview of Model architectures (a) Single-column architecture: a method for training the model using either EEG or GSR data (b) Multiple-column architecture: the presented model comprises two single-column models, each equipped with dedicated model backbones(either CNN or transformer-based) for their respective modalities. GSR and EEG features are extracted independently from each model backbone and then concatenated.

model initialized with the same learning rate of 0.01 as the single model or initialized with ImageNet weights, it generally showed slightly better performance (data not shown). Regarding the training duration, training the single-column model across 5 folds and 30 epochs took approximately 1 hour. In contrast, the multi-column model required about 1.5 hours to complete the same number of epochs.

## 2.5 Evaluation of the model and experimental settings

To optimize model parameters, a five-fold cross-validation approach was employed. To prevent data from the same subject from mixing in the training and testing sets, we partitioned the dataset into five subsets on a subject-wise basis. Each subject's data was exclusively assigned to one of these subsets, with four subsets allocated for model training and the remaining one reserved for validation. This method ensures no overlap between training and testing data, thereby mitigating potential overestimation of model performance. Regarding model evaluation, the area under the receiver operating characteristic curve (AUROC) was computed as the performance metric for each model. All models were trained using PyTorch, with training conducted on a Quadro RTX 8000 GPU. The experiments were conducted on a 64-bit computer processor housing an Intel® Xeon® Gold 6226R CPU @ 2.90GHz with 16 cores.

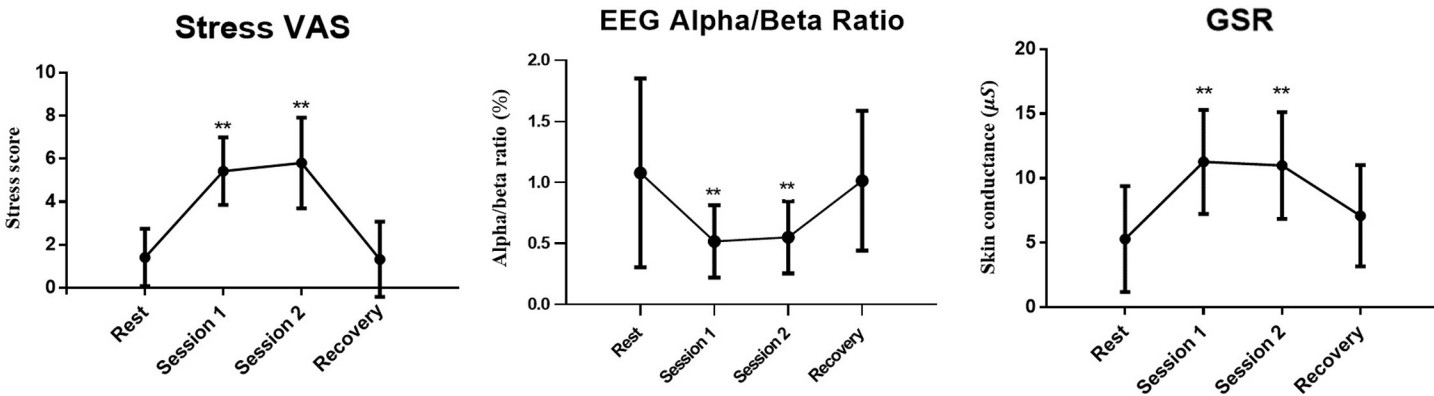

**Fig 3. Experiment result: An analysis of the segment-specific scores and the averaged values of respective physiological signals for rest, Session 1, Session 2, and recovery, as indicated in Fig 4.** (paired t-test, *p < 0.05, **p < 0.005).

## 3. Result

### 3.1 Biosignal characteristics on each session of experiments

Stress levels were assessed using the visual analogue scale (VAS) score, as depicted in Fig 3. The granularity of the labeling was achieved using a 10-cm visual analogue scale where participants marked their level of stress. These marks were then measured with a ruler to millimeter precision, so, for example, a mark at 4.2 cm on the scale would score a 4.2. Measurements were taken a total of four times, before and after each session, as shown in Fig 4. During the resting period, the VAS score increased from 1.4 ± 1.3 to 5.4 ± 1.6 and 5.8 ± 2.1 in session 1 and session 2, respectively. To gauge the impact of stress on EEG, we computed the alpha (8 – 13Hz)/ beta (13 – 30Hz) ratio in the frequency domain. During the resting period, the ratio measured 1.1 ± 0.8; however, it decreased to 0.52 ± 0.3 and 0.6 ± 0.3 in session 1 and session 2, respectively. Similarly, GSR measurements demonstrated elevated stress levels transitioning from the rest period (5.3 ± 4.1) to session 1 (11.3 ± 4.0) and session 2 (11.0 ± 4.1). These findings collectively indicate the effective induction of stress responses in participants through the experimental stress protocol, substantiated by discernible alterations in both EEG and GSR measurements.

### 3.2 Image-wise classification

The proposed models were trained and validated using 551 images for Normal and 1529 images for Stress, with performance assessment conducted through five-fold cross-validation

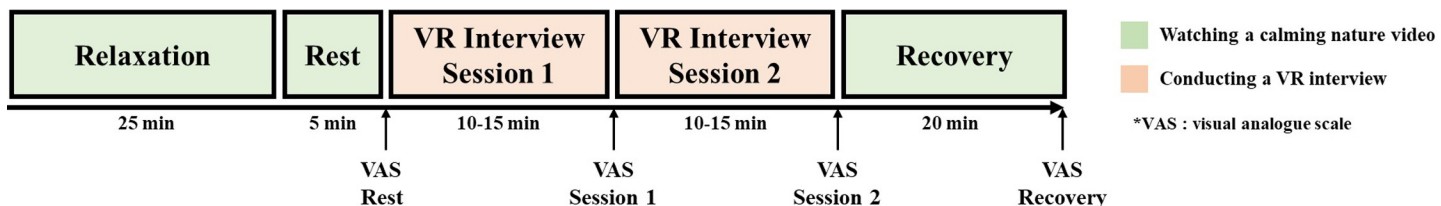

**Fig 4. Experimental sequence: The experimental sequence, as depicted in Fig 1, began with a 30-minute period of rest and relaxation, followed by two 15-minute sessions of VR interviews intended to induce stress.** The final interview session was followed by a 20-minute recovery interval. During the rest and recovery phases, participants utilized VR equipment (MintPot Co., Ltd.) to view serene natural landscapes. The VR interviews were conducted using the "God of interview" program (MintPot Co., Ltd), simulating a virtual mock interview experience. Stress levels were assessed using a stress visual analogue scale (VAS) questionnaire administered both before and after each session.

as shown in Table 1. The experimental outcomes revealed that within the category of single-column models, ResNet-152 achieved the highest AUROC value for GSR, while Vision Transformer demonstrated the highest AUROC value for EEG. Specifically, ResNet-152 exhibited an AUROC value of 0.944 for Stress and Normal classification using GSR, while the Vision Transformer yielded an AUROC value of 0.886 for EEG. For the multiple-column structure, which combined EEG and GSR features and used ResNet-50 as the foundation, the highest AUROC value of 0.954 for Stress and Normal classification was attained. This suggests that the model effectively extracts complementary features from each modality, potentially capturing their interconnected nature.

### 3.3 Visualizing stress detection results

We employed the ResNet50 model to visualize stress detection outcomes across the entire duration of an experiment for a given subject. As Fig 5 depicts, the subject gradually transitions to a state of calmness during the relaxation phase, resulting in a decline in stress probability over time. Subsequently, in the rest state, stress probability consistently maintains a low level. In contrast, during the VR job interview stress session, the stress probability displays an ascent, followed by a gradual decrease during the subsequent recovery phase. This distinct pattern exemplifies accurately classified observations, underscoring the model's adeptness in capturing discernible stress fluctuations across distinct stages.

In Fig 6, results from another test involving a different subject are visualized. As shown in Fig 6(A), it can be observed that GSR signals in the resting and stress periods do not appear as clearly distinguishable compared to the previous example in Fig 5(A). Certain individuals may exhibit physiological signals, such as GSR or EEG, that don't exhibit a distinct difference between resting and stressful situations. Such individuals might encounter challenges in

**Table 1. The average evaluation metrics of each model for stress detection with 5-fold cross validation.**

| Models | Model size (MB) | Input data | AUC | Accuracy | Recall | Precision | F1 score |
|---|---|---|---|---|---|---|---|
| Inception-v3 | 83 | GSR (Single column) | 0.932 (±0.033) | 0.873 (±0.030) | 0.913 (±0.064) | 0.921 (±0.060) | 0.913 (±0.021) |
| | | EEG (Single column) | 0.878 (±0.038) | 0.778 (±0.041) | 0.794 (±0.102) | 0.899 (±0.047) | 0.837 (±0.037) |
| | | GSR, EEG (Multiple column) | 0.943 (±0.013) | 0.886 (±0.051) | 0.921 (±0.099) | 0.923 (±0.029) | 0.918 (±0.047) |
| ResNet-50 | 90 | GSR | 0.929 (±0.029) | 0.860 (±0.043) | 0.892 (±0.077) | 0.919 (±0.042) | 0.902 (±0.034) |
| | | EEG | 0.847 (±0.037) | 0.701 (±0.099) | 0.686 (±0.218) | 0.895 (±0.057) | 0.748 (±0.139) |
| | | GSR, EEG | **0.954 (±0.018)** | 0.865 (±0.072) | 0.887 (±0.143) | 0.928 (±0.030) | 0.898 (±0.070) |
| ResNet-152 | 222 | GSR | 0.944 (±0.027) | 0.865 (±0.063) | 0.860 (±0.078) | 0.949 (±0.023) | 0.901 (±0.049) |
| | | EEG | 0.868 (±0.029) | 0.760 (±0.070) | 0.764 (±0.161) | 0.902 (±0.046) | 0.814 (±0.072) |
| | | GSR, EEG | 0.946 (±0.059) | 0.889 (±0.059) | 0.908 (±0.115) | 0.938 (±0.023) | 0.918 (±0.058) |
| DenseNet-161 | 102 | GSR | 0.924 (±0.057) | 0.849 (±0.102) | 0.841 (±0.136) | 0.939 (±0.031) | 0.883 (±0.093) |
| | | EEG | 0.861 (±0.033) | 0.805 (±0.030) | 0.830 (±0.045) | 0.896 (±0.015) | 0.861 (±0.023) |
| | | GSR, EEG | 0.907 (±0.030) | 0.784 (±0.058) | 0.815 (±0.132) | 0.891 (±0.065) | 0.842 (±0.062) |
| EfficientNet–b0 | 16 | GSR | 0.931 (±0.025) | 0.896 (±0.030) | 0.950 (±0.045) | 0.913 (±0.037) | 0.930 (±0.021) |
| | | EEG | 0.863 (±0.051) | 0.813 (±0.041) | 0.857 (±0.077) | 0.887 (±0.043) | 0.868 (±0.038) |
| | | GSR, EEG | 0.948 (±0.014) | 0.867 (±0.093) | 0.874 (±0.167) | 0.941 (±0.028) | 0.894 (±0.094) |
| Vision Transformer | 328 | GSR | 0.940 (±0.014) | 0.866 (±0.014) | 0.937 (±0.054) | 0.892 (±0.054) | 0.911 (±0.015) |
| | | EEG | 0.886 (±0.069) | 0.758 (±0.109) | 0.782 (±0.229) | 0.895 (±0.087) | 0.803 (±0.144) |
| | | GSR, EEG | 0.945 (±0.033) | 0.882 (±0.054) | 0.912 (±0.066) | 0.927 (±0.020) | 0.918 (±0.038) |

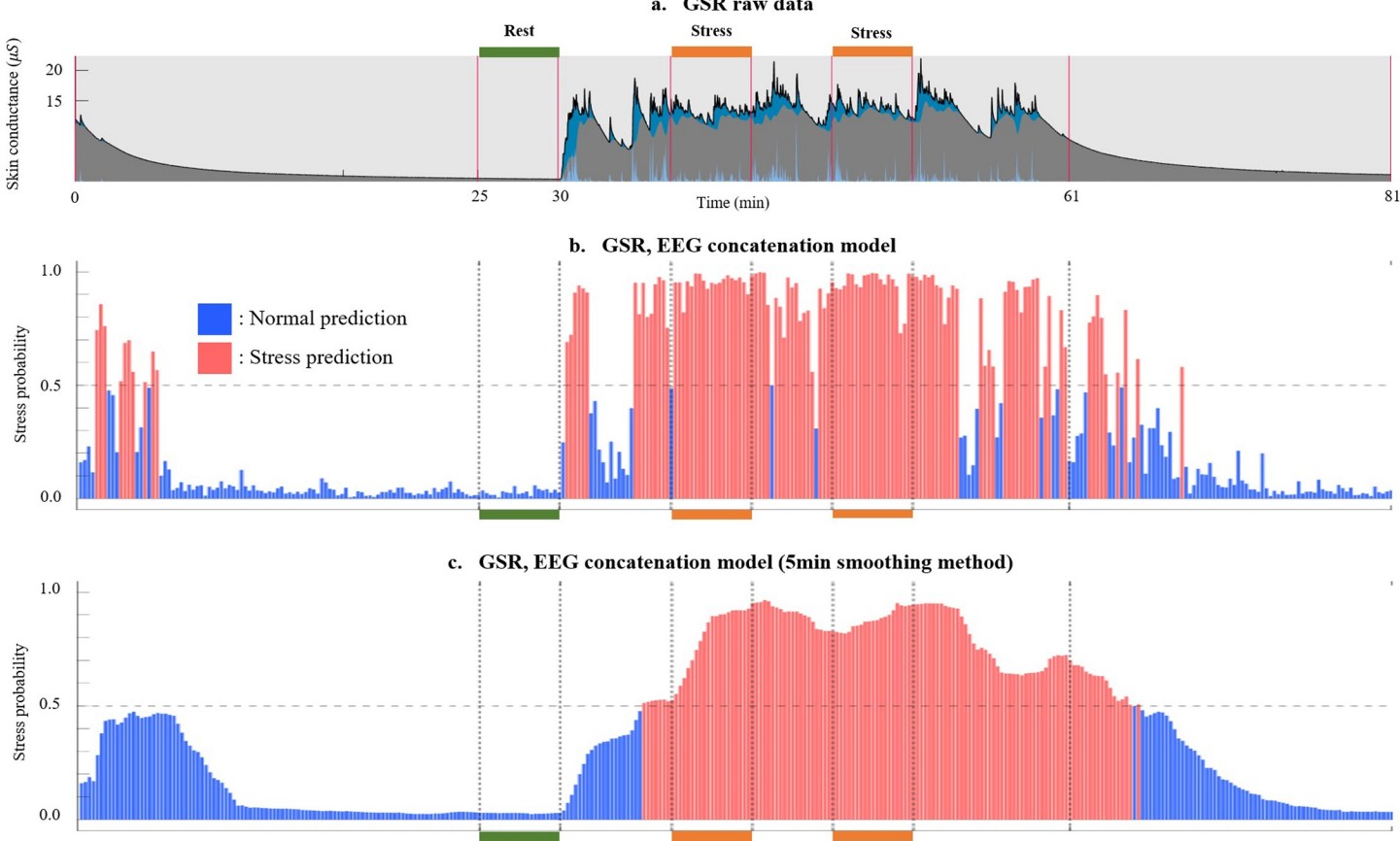

**Fig 5.** AI predicted outcome analysis: Each graph's x-axis signifies time, with green and orange markings above or below the x-axis corresponding to actual normal and stress data used for training the deep learning model (a) This denotes the GSR raw data, where the blue segment indicates the phasic component, and the gray signifies the tonic component, as determined using LEDALAB tool. (b) Analyzing the stress probability throughout the entire experimental process was conducted using a model trained with the multiple-column architecture (c) The outcomes after applying a smoothing technique, which is a post-processing method that calculates the average probability of the 5-minute window to mitigate abrupt misclassifications. In each graph, the color of the bars indicates whether the probability exceeds 0.5 (depicted in red) or falls below 0.5 (shown in blue).

remaining calm even during a resting period within these experiments. Application of the concatenation method, as depicted in Fig 6(B), makes stress state detection difficult.

## 3.4 Image classification models for stress detection with smoothing method

We proceeded to assess the performance of stress detection with the incorporation of the smoothing technique across varying time windows—1, 3, and 5 minutes as shown in Table 2. The reported accuracies were derived by calculating a moving average of stress probability within the designated time window. Among the models scrutinized, ResNet-50 and Inception-v3 consistently showcased the highest average AUROC accuracy across all time windows, yielding AUROC values of 0.973 (±0.021), 0.990 (±0.020), and 0.993 (±0.017), respectively. Notably, EfficientNet-b0 also demonstrated remarkable performance, recording average AUROC values of 0.954 (±0.017), 0.970 (±0.028), and 0.981 (±0.024). In contrast, DenseNet-161 exhibited relatively lower average AUROC values across all time windows, registering results of 0.918 (±0.034), 0.935 (±0.033), and 0.939 (±0.044). In summary, Inception-v3, ResNet-50, and EfficientNet-b0 emerged as the top-performing models in terms of AUROC accuracy. These findings underscore the critical role of selecting an appropriate model for

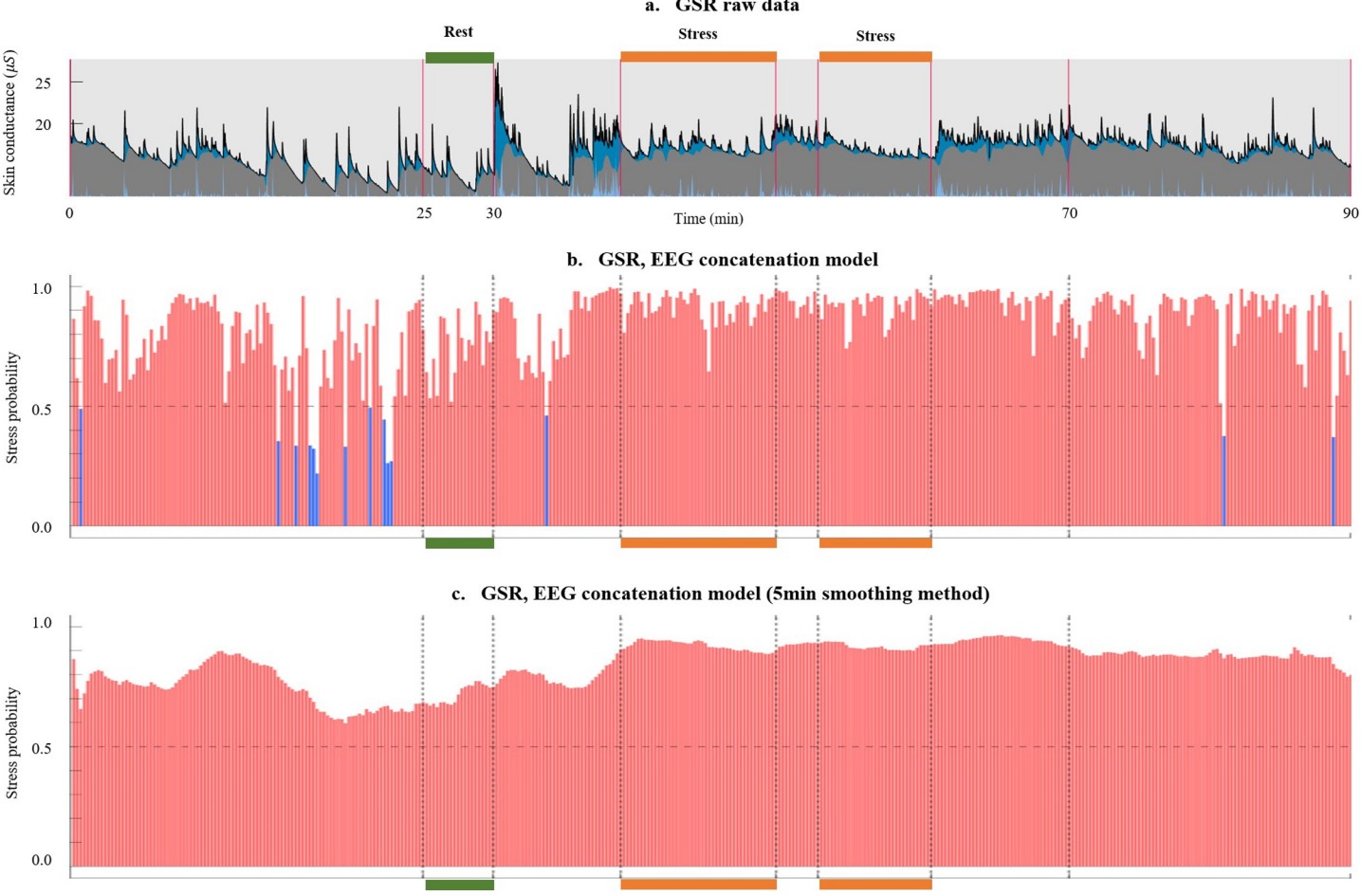

**Fig 6. AI variable outcome analysis: A comprehensive outcome.** Fig 6(a)-6(c) correspond to Fig 5, sharing identical content.

stress classification, accounting for both prediction accuracy and the desired prediction time window.

### 3.5 Grad-CAM analysis of the stress classification results

The Grad-CAM (Gradient-weighted Class Activation Mapping) technique [34] is commonly employed in deep learning models to visualize regions of importance during the classification process. In our study, we aimed to discern which frequency bands of biosignals play a crucial role in classifying normal and stress states. After applying the Grad-CAM analysis, as

**Table 2. Stress detection results after applying the smoothing technique with various time windows.**

| Models | 1 min | 3 min | 5 min |
| --- | --- | --- | --- |
| Inception-v3 | 0.961 (±0.022) | 0.969 (±0.021) | 0.992 (±0.009) |
| ResNet-50 | 0.973 (±0.021) | 0.990 (±0.020) | **0.993 (±0.017)** |
| ResNet-152 | 0.959 (±0.029) | 0.970 (±0.040) | 0.975 (±0.017) |
| DenseNet-161 | 0.918 (±0.034) | 0.935 (±0.033) | 0.939 (±0.044) |
| EfficientNet–b0 | 0.954 (±0.017) | 0.970 (±0.028) | 0.981 (±0.024) |
| Vision Transformer | 0.940 (±0.049) | 0.960 (±0.045) | 0.967 (±0.045) |

illustrated in Fig 7, we computed the mean highlighted frequency for each class within each signal modality. For GSR, the Grad-CAM analysis highlighted approximately 8.86 ± 0.93Hz and 9.19 ± 1.41Hz for normal and stress states, respectively. Additionally, an analogous analysis was conducted for the EEG signal modality, revealing a mean highlighted frequency of 15.86 ± 3.28Hz for the normal state and 16.50 ± 2.96Hz for the stress state.

## 4. Discussion

Informed by the widely recognized trier social stress test [33], a paradigm known for its ability to induce stress through traditional face-to-face interviews, we adapted a VR-based interview approach with similarities to the former. Subsequently, by administering stress VAS questionnaires before and after the interview sessions, we received reports indicating the induction of stress. Moreover, we were able to confirm significant alterations in physiological signals, which are indicative of stress-related responses. Thus, this investigation establishes that socially induced stress, provoked through VR-based simulated interviews, results in distinguishable modifications in physiological signals.

Extensive prior research has consistently reported the influence of stress on both the tonic and phasic components of GSR [34]. GSR encapsulates not only gradual shifts (tonic component) represented by mean values but also incorporates information about rapid or phasic signal variations. In our study, we observed a statistically significant increase in the tonic GSR component in response to social stress. Previous studies have explored the stress-EEG nexus; however, many focused on forehead measurements or employed multi-channel EEG cap systems [35–37], limiting the acquisition of real-life data. Nevertheless, we detected a noteworthy rise in the alpha/beta band ratio of EEG when measured behind the ear. These analytical outcomes underscore the viability of using physiological signals such as GSR and EEG to assess stress induced by interviews, akin to other stress-inducing factors. Although the alpha/beta ratio isn't a frequently utilized EEG analysis feature, its physiological significance calls for additional exploration [38].

We compared the stress classification performance of diverse deep learning models, including CNNs and a Vision Transformer, using GSR and EEG signals. For the single-column models, utilizing 30-second physiological signal data, we achieved AUROC results

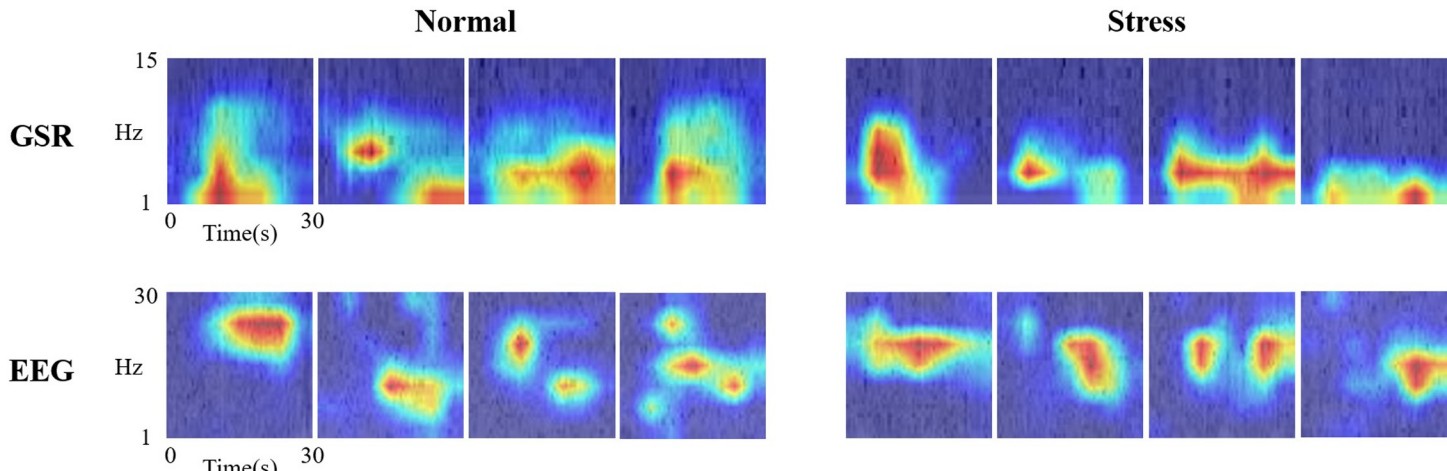

**Fig 7. Grad-CAM visualizations of GSR and EEG spectrograms associated with stress responses.** The x-axis in the images represents time intervals of 30 seconds, while the y-axis signifies frequency ranges: 1-15Hz for GSR and 1-30Hz for EEG.

exceeding 93% for GSR and surpassing 85% for EEG. The multiple-column models exhibited superior performance compared to their single-column counterparts. Consequently, our models highlight the potential for detecting stress states using these physiological signals, with the multiple-column configurations demonstrating enhanced performance compared to single-column models that utilize only one modality. The augmented AUROC obtained from concurrent GSR (measured via the autonomic nervous system) and EEG (measured via the central nervous system) might imply that stress impacts both the autonomic and central nervous systems, aligning with established knowledge [39]. Additionally, while our model's present state involves training on the complete dataset of 29 subjects, future iterations could entail personalized adjustments, potentially enhancing model capacity, particularly for subjects resembling the example depicted in Fig 6.

Furthermore, we introduced a smoothing technique to facilitate the tracking of stress level changes, mitigating abrupt misclassification. This approach enhances error reduction while rendering the comprehensive stress level fluctuations more comprehensible. Although a-5minute window might seem excessive for 10-15minute sessions, our testing across 1, 3, and 5-minute intervals demonstrated that the 5-minute window effectively captures sustained stress patterns and minimizes misclassification. We recommend further studies with longer sessions to refine this method. Longer-term data validation is warranted, the prospective amalgamation of behind-the-ear EEG and GSR with wearable devices holds promise for aiding stress monitoring [40]. In addition, although this study exclusively utilized spectrograms for training our models, future investigations could also explore employing the continuous wavelet transform (CWT) to potentially enhance our ability to analyze and classify complex signal patterns.

The method of analyzing biosignals in the frequency domain, as well as the time domain, has been widely used in previous research. Our study was informed by the knowledge that GSR signals change more rapidly under stress conditions and that, as seen in Fig 3, there are variations in the alpha/beta ratio. We anticipated differences in the frequency domain as a result. Based on these insights, we aimed to investigate whether our AI model would predominantly consider these differences in frequency power during its learning process. Despite applying Grad-CAM analysis, the results, as observed, did not show significant differences in the Hz regions monitored before and after stress. This observation suggests that prioritizing overall information rather than focusing on individual frequency bands may be crucial for the accurate classification of normal and stress states.

GSR measurements were taken on the index and middle fingers of the left hand, while EEG was measured behind the ear. For practical applications of stress analysis through physiological signals in daily life, wearable and socially acceptable measurement devices are essential. Our unobtrusive data acquisition device, positioned behind the ear, offers discretion from frontal view and aligns well with real-life scenarios. In the future, the concurrent development of devices capable of measuring both GSR and EEG from the auricular region holds the potential to amplify the efficacy of stress monitoring in real-world settings. In conclusion, this study achieved precise stress analysis utilizing a multiple-column network, leveraging data from behind-the-ear EEG measurements along with GSR measurements from the fingers. The resulting model enables stress analysis at 30-second intervals, facilitating real-time detection of individual stressors in everyday life without the need for complex biosignal measurement systems. This research holds the potential to enhance mental health monitoring and treatment.

## Author Contributions

**Methodology:** Hun-gyeom Kim, Solwoong Song.

**Project administration:** Baek Hwan Cho, Dong Pyo Jang.

**Software:** Hun-gyeom Kim, Solwoong Song.

**Supervision:** Baek Hwan Cho, Dong Pyo Jang.

**Visualization:** Hun-gyeom Kim, Solwoong Song.

**Writing – original draft:** Hun-gyeom Kim, Solwoong Song.

**Writing – review & editing:** Baek Hwan Cho, Dong Pyo Jang.

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
