## [Decision Letter · Decision Letter 0]

8 Apr 2024

PONE-D-24-09412Deep Learning-based Stress Detection for Daily Life Use Using Single-Channel EEG and GSR in a Virtual Reality Interview ParadigmPLOS ONE

Dear Dr. Cho,

Thank you for submitting your manuscript to PLOS ONE. After careful consideration, we feel that it has merit but does not fully meet PLOS ONE’s publication criteria as it currently stands. Therefore, we invite you to submit a revised version of the manuscript that addresses the points raised during the review process.

We look forward to receiving your revised manuscript.

Kind regards,

Xiaohui Zhang

Academic Editor

PLOS ONE

Journal Requirements:

"This work was supported by the Brain Convergence Research Program of the National Research Foundation (NRF) funded by the Korean government (MSIT) (No. NRF-2021M3E5D2A01022391)."

4. We note that Figures 1 and 2 in your submission contain copyrighted images. All PLOS content is published under the Creative Commons Attribution License (CC BY 4.0), which means that the manuscript, images, and Supporting Information files will be freely available online, and any third party is permitted to access, download, copy, distribute, and use these materials in any way, even commercially, with proper attribution. For more information, see our copyright guidelines: http://journals.plos.org/plosone/s/licenses-and-copyright.

a. You may seek permission from the original copyright holder of Figures 1 and 2 to publish the content specifically under the CC BY 4.0 license. 

Reviewers' comments:

Reviewer's Responses to Questions

**Comments to the Author**

1. Is the manuscript technically sound, and do the data support the conclusions?

Reviewer #1: Yes

Reviewer #2: Yes

2. Has the statistical analysis been performed appropriately and rigorously? 

Reviewer #1: N/A

Reviewer #2: No

3. Have the authors made all data underlying the findings in their manuscript fully available?

Reviewer #1: No

Reviewer #2: No

4. Is the manuscript presented in an intelligible fashion and written in standard English?

Reviewer #1: Yes

Reviewer #2: Yes

5. Review Comments to the Author

Reviewer #1: This manuscript describes the use of different deep learning models to identify stress and clam states using EEG and GSR signals. The manuscript is well written but the authors should address the following questions and comments?

1. Why is the learning rate of the multi-column model 10 times smaller than the single column models?

2. In the current setting, the multi-column model was initialized from the optimized single column models. Does the performance vary significantly if the multi-column model was instead initialized with the same Image-Net weights and trained using the same learning rate as the single column models?

3. The authors used spectrograms to convert time-series input signal to a 2D image for the CNN training. Has the authors considered any other time-frequency analysis methods, especially the continuous wavelet transform (CWT)? CWT has seen great success in preparing 2D data for CNN-classifiers in many other fields, for example:

https://doi.org/10.1121/1.3384871

https://doi.org/10.1115/IMECE2023-112407

4. The authors used a 5min averaging window to smooth out the predictions. Is this window length too much, considering that the interview session is only 10-15mins?

5. The training time of each model should be reported.

Reviewer #2: The authors stimulated people's stress state through virtual interviews, assessed people's stress state using questionnaires, and modeled people's stress state through EEG signals and GSR signals. Demonstrated that neural networks can assess human stress levels from both models. Here are comments:

1. By using VAS questionnaire as the evaluation method. How often did the subjects be test by the VAS? What is the granularity of the labeling?

2. GSR and EEG are used as features. How are they combined in the networks? Is normalization needed?

3. The last sentence of Section 2.2 is missing a period, and it seems like the sentence isn't complete.

4. Why are there fewer normal samples compared to stress samples? It should be easier to collect normal samples.

5. The dataset partitioning method should be introduced. If a patient appears in both the training and testing sets, it could introduce bias into the results.

6. In Section 2.4, all models were pretrained on ImageNet. Why was this done? There's a significant difference between spectrogram and ImageNet images.

7. Section 2.4 and Figure 3 introduced the methods used, including the development of the multiple-column CNN model. However, the Vision Transformer does not belong to CNN. The author should clarify how it was incorporated.

8. Are hyperparameters such as batch size and learning rate the same across all models?

9. Why was a 5-minute interval chosen for smoothing in Figure 5?

10. AUC alone is not sufficient. Other metrics should be provided for comparison.

11. What is the purpose of Figure 7? What information can we gain from it? GradCAM operates in a two-dimensional space, but the original EEG and GSR signals are one-dimensional. Is this approach reasonable?

12. Data and code should be made publicly available.

6. PLOS authors have the option to publish the peer review history of their article (what does this mean?). If published, this will include your full peer review and any attached files.

Reviewer #1: No

Reviewer #2: No

---

## [Author Response · Author response to Decision Letter 0]

22 May 2024

May 16, 2024

Manuscript ID PONE-D-24-09412

Title: Deep Learning-based Stress Detection for Daily Life Use Using Single-Channel EEG and GSR in a Virtual Reality Interview Paradigm

Dear Editor,

We thank you and the reviewers for your interest in our paper and for your comments, which have helped us improve the manuscript substantially. We have revised the paper in response to these comments, and we are glad to submit the revised manuscript for your consideration. We hope the revised manuscript will meet the requirements of your journal for publication. 

Please do not hesitate to contact me if you have additional questions or issues regarding this paper.

We look forward to your final decision on this manuscript.

With best wishes,

Baek Hwan Cho, Ph.D.

Department of Biomedical Informatics, CHA University School of Medicine, CHA

University, Seongnam, Republic of Korea

Institute of Biomedical Informatics, School of Medicine, CHA University, Seongnam,

Republic of Korea

---

## [Decision Letter · Decision Letter 1]

6 Jun 2024

Deep Learning-based Stress Detection for Daily Life Use Using Single-Channel EEG and GSR in a Virtual Reality Interview Paradigm

PONE-D-24-09412R1

Dear Dr. Cho,

We’re pleased to inform you that your manuscript has been judged scientifically suitable for publication and will be formally accepted for publication once it meets all outstanding technical requirements.

Kind regards,

Xiaohui Zhang

Academic Editor

PLOS ONE

Additional Editor Comments (optional):

Thank the authors for addressing reviewers' comments.

Reviewers' comments:

Reviewer's Responses to Questions

**Comments to the Author**

1. If the authors have adequately addressed your comments raised in a previous round of review and you feel that this manuscript is now acceptable for publication, you may indicate that here to bypass the “Comments to the Author” section, enter your conflict of interest statement in the “Confidential to Editor” section, and submit your "Accept" recommendation.

Reviewer #1: All comments have been addressed

Reviewer #2: All comments have been addressed

2. Is the manuscript technically sound, and do the data support the conclusions?

Reviewer #1: Yes

Reviewer #2: Yes

3. Has the statistical analysis been performed appropriately and rigorously? 

Reviewer #1: Yes

Reviewer #2: Yes

4. Have the authors made all data underlying the findings in their manuscript fully available?

Reviewer #1: Yes

Reviewer #2: Yes

5. Is the manuscript presented in an intelligible fashion and written in standard English?

Reviewer #1: Yes

Reviewer #2: Yes

6. Review Comments to the Author

Reviewer #1: The authors have addressed the reviewer's comment sufficiently. This manuscript can be recommended for publication.

Reviewer #2: Thanks for the author's reply. I think they have answered my questions and made corresponding changes.

7. PLOS authors have the option to publish the peer review history of their article (what does this mean?). If published, this will include your full peer review and any attached files.

Reviewer #1: No

Reviewer #2: No

---

## [Editor Report · Acceptance letter]

23 Jun 2024

PONE-D-24-09412R1 

PLOS ONE

Dear Dr. Cho, 

I'm pleased to inform you that your manuscript has been deemed suitable for publication in PLOS ONE. Congratulations! Your manuscript is now being handed over to our production team.

Kind regards, 

on behalf of

Dr. Xiaohui Zhang 

Academic Editor

PLOS ONE